# Efficacy and Safety of Bojungikgi-Tang for Persistent Allergic Rhinitis: A Randomized, Double-Blinded, Placebo-Controlled, Phase II Trial

**DOI:** 10.3390/healthcare12101017

**Published:** 2024-05-14

**Authors:** Su Won Lee, Seong-Cheon Woo, Yee Ran Lyu, Won-Kyung Yang, Seung-Hyung Kim, Je Hyun Kim, Si Yeon Kim, Weechang Kang, In Chul Jung, Taesoo Kim, Yang Chun Park

**Affiliations:** 1Division of Respiratory Medicine, Department of Internal Medicine, College of Korean Medicine, Daejeon University, Daejeon 34520, Republic of Korea; tndnjs3325@daum.net (S.W.L.); wsc672@naver.com (S.-C.W.); ywks1220@dju.kr (W.-K.Y.); 2Korea Institute of Oriental Medicine, Daejeon 34054, Republic of Korea; onedoctor2ran@kiom.re.kr; 3Institute of Traditional Medicine and Bioscience, Daejeon University, Daejeon 34520, Republic of Korea; sksh518@dju.kr; 4Clinical Trial Center, Daejeon Korean Medicine Hospital of Daejeon University, Daejeon 35235, Republic of Korea; kimjehyun0717@naver.com (J.H.K.); ksymylove89@naver.com (S.Y.K.); npjeong@dju.kr (I.C.J.); 5Department of Statistics, Graduate School, Daejeon University, Daejeon 34520, Republic of Korea; weechang@dju.kr; 6Department of Neuropsychology, College of Korean Medicine, Daejeon University, Daejeon 34520, Republic of Korea

**Keywords:** allergic rhinitis, Bojungikgi-tang, Hochu-ekki-to, Bu-Zhong-Yi-Qi-Tang, herbal medicine, randomized controlled trial

## Abstract

Conventional treatments for allergic rhinitis (AR) exhibit insufficiency and long-term use-related side effects. Considering the reported anti-inflammatory and immunoregulatory effects of Bojungikgi-tang (BJIGT), we aimed to assess its efficacy on persistent AR (PAR). Patients with PAR were randomly assigned in a 1:1:1 ratio into high-dose BJIGT, standard-dose BJIGT, and placebo groups, followed by 1-week run-in and 4-week treatment periods. The primary outcome included the mean change in Total Nasal Symptom Score (TNSS), with secondary outcomes encompassing the Korean Allergic Rhinitis-Specific Quality of Life Questionnaire, biomarkers, overall assessment, TNSS by AR pattern identification, and the Sasang constitution. The mean TNSS change was more improved in the BJIGT group than in the placebo group; however, no statistically significant differences were observed. Additional interaction effect analysis revealed a statistically significant improvement in the high-dose BJIGT group compared with the placebo group from weeks 1–2 to weeks 3–4. Regarding secondary outcomes, the BJIGT group exhibited similar or improved results compared with the placebo group, showing no statistically significant differences. No serious adverse effects or clinically significant changes in safety assessments were observed. Given that this study validated clinical improvement and safety, it serves as potential groundwork for pertinent future studies.

## 1. Introduction

Allergic rhinitis (AR) is caused by a gamma globulin E (IgE)-mediated immune response, resulting in inflammation-induced symptomatic manifestations such as sneezing, itching, nasal congestion, and rhinorrhea [1]. AR is a prevalent condition, affecting 10% to 40% of the global population [2]. In South Korea, the estimated prevalence of AR between 2016 and 2017 was 27.6% in school-age children and 17.1% in adults, with a significant upward trend over the last decade [3]. AR can be divided into intermittent AR (IAR) and persistent AR (PAR) based on symptom duration [2]. Although not life-threatening, the symptoms of AR significantly affect daily life, and its high prevalence contributes to a substantial social and economic burden [4]. Furthermore, untreated AR poses a risk for concomitant diseases such as sinusitis, asthma, and otitis media [5].

Treatment strategies for AR typically include allergen avoidance, pharmacotherapy, immunotherapy, and patient education [6]. Among these, the primary treatment encompasses medication for relieving symptoms owing to the difficulty associated with complete allergen avoidance in daily life. Oral and topical antihistamines, intranasal corticosteroids, and decongestants are also employed either solely or combined, depending on the situation of the patient [7]. However, these treatments do not afford complete control over AR symptoms, and long-term use for recurrent AR is limited by several adverse events (AEs) [8]. For instance, intranasal corticosteroids can produce throat discomfort, and intranasal cromolyn can produce nosebleeds [9].

Complementary medicine has gained widespread application for AR treatment, with several herbal medicines demonstrating efficacy in many trials [10,11,12]. Bojungikgi-tang (BJIGT), also known as Hochu-ekki-to or Bu-Zhong-Yi-Qi-Tang, is an herbal medicine that has been used for centuries in traditional medicine. The nomenclature of BJIGT, “tonify the middle and augment qi”, indicates its role in strengthening the immune and digestive systems and addressing debilitation resulting from chronic diseases. Previous studies have demonstrated the suppressive effects of BJIGT on eosinophil levels, as well as its protective effects on the nasal mucosa in ovalbumin-inhalation rat models of AR [13]. Additionally, immunomodulatory effects have been observed, including reductions in the levels of IgE, eotaxin, and eosinophils and the downregulation of Th2 responses in an allergic asthma model [14]. In addition to observations in preclinical trials, symptoms of the nose in PAR patients have been notably reduced in BJIGT groups, and the anti-inflammatory effects of BJIGT have been verified in PAR patients [15,16]. Moreover, survey studies have shown that BJIGT is one of the most used insured herbal medicines for AR in South Korea [17,18]. The safety of BJIGT has also been verified in a toxicity trial [19]. However, a randomized controlled trial investigating the efficacy of BJIGT for human patients with AR in South Korea has not been conducted to date.

Therefore, we designed a randomized, placebo-controlled, double-blind, phase II trial—utilizing two BJIGT doses and a placebo—to investigate the efficacy and safety of BJIGT for PAR and determine optimal dosage parameters, targeting PAR with a long symptom duration rather than IAR with a short symptom duration. The findings of this phase II trial validate the safety and efficacy of BJIGT, providing valuable data for informing future studies in this field.

## 2. Materials and Methods

The study protocol was approved by the Korean Ministry of Food and Drug Safety (MFDS) through an investigational new drug (IND) application in May 2021. Furthermore, approval was granted by the Institutional Review Board of Daejeon Korean Medicine Hospital, Daejeon University (approval number: DJDSKH-20-DR-13) in June 2021. This study was registered at the National Clinical Trial Registry Clinical Research Information Service (https://cris.nih.go.kr) accessed on 26 July 2021 under the identifier KCT0006616.

### 2.1. Design and Procedures

This study was a randomized controlled trial on the efficacy and safety of BJIGT in the treatment of PAR. The trial was a double-blind trial and was performed at Daejeon Korean Medicine Hospital of Daejeon University, South Korea, following a previously published protocol [20].

The patients were subjected to comprehensive screening encompassing chest X-ray, laboratory tests, electrocardiogram, and a multiple allergen simultaneous test (MAST) to rule out the presence of other diseases. Eligible patients who met the inclusion criteria were randomly assigned in a 1:1:1 ratio into three groups: high-dose BJIGT (BJIGT, 15 g/day), standard-dose BJIGT (BJIGT, 7.5 g/day), and placebo groups. The trial comprised run-in (1 week) and treatment periods (4 weeks). Participants were administered the investigational products thrice daily for 4 weeks, with scheduled visits every 2 weeks, from weeks 0 to 4. Outcome measures and safety assessments, including AEs, vital signs, and laboratory examinations, were conducted in the treatment.

### 2.2. Participants

The trial encompassed patients (aged 19–65 years) with a *PAR* (more than 4 days a week and 4 or more consecutive weeks) history of minimum 2 years before enrolment and a positive reaction to perennial allergens in allergy skin tests, MAST, or ImmunoCAP conducted within the preceding 12 months. The average daily reflective Total Nasal Symptom Score (r-TNSS) during the run-in period was stipulated to be at least 5 points. Exclusion criteria included non-AR, concurrent asthma, and previous usage of long-acting antihistamines. A detailed list of inclusion and exclusion criteria is provided in the study protocol [20].

### 2.3. Sample Size

The determination of the sample size was based on the primary outcome, namely, the mean change in TNSS from baseline to the end of the medication period among patients with AR. A previous study involving other herbal medications versus placebo reported a TNSS difference of 0.97 and a standard deviation of 1.3 [11]. Adopting a significance level of 5% (α) and a power of 80% (1–β), the calculated sample size was 29 patients per group. Considering a dropout rate of 20%, 35 participants were required per group (a total of 105 subjects). This approach aligns with the variability observed in sample sizes across previous phase II trials, ranging from 20 to 42 per group [21,22,23].

### 2.4. Randomization and Blinding

Randomization was performed by an independent statistician using a random number generator in the SAS version 9 statistical software (SAS Institute, Cary, NC, USA). Allocation was implemented by the manufacturers, who collectively labeled the packages of the investigational products with the identification codes of the participants using generated random numbers. The management pharmacist provided the participants with a labeled drug following the identification code of the participant. The manufacturers and statistician had exclusive access to the random numbers, and only the identification code was applied to recognize which drug to administer to which patient. This study was a double-blind trial in which neither the participants nor investigators were aware of the group assignment during the study period.

### 2.5. Interventions

During the run-in period, the participants were administered a placebo, followed by a 4-week treatment period where they were administered BJIGT (high-dose or standard-dose) or placebo, according to their assigned group. BJIGT and placebo granules were encapsulated within aluminum bags, with participants instructed to administer two bags (5.0 g) thrice daily. The dosage allocation for the high- and standard-dose BJIGT groups, established at 15 g/day and 7.5 g/day, respectively, was determined by the Korean MFDS alongside pharmacologically active dose determinations in expectorant effective tests [24].

BJIGT contains ten raw medicinal herbs, and the ingredients and proportions are shown in Table 1. The above herbs were extracted with boiling water to make a soft extract, and by adding excipients, BJIGT was manufactured as 2.5 g of dried granules. By contrast, the placebo formulation lacked these active ingredients and comprised lactose, cornstarch, and caramel coloring, among other inert components. Both granules shared identical weights, shapes, and a gray-brown color. Hankookshinyak Corporation (Nonsan, Republic of Korea) manufactured the BJIGT and placebo according to Korean Good Manufacturing Practice standards.

### 2.6. Outcomes

#### 2.6.1. Primary Outcome

The primary outcome encompassed the mean change in TNSS before and after treatment across the three groups. TNSS, widely applied for evaluating the symptoms of patients with AR, comprises four symptom categories (rhinorrhea, nasal congestion, nasal itching, and sneezing), each rated on a 4-point severity scale (0 = no symptoms, 1 = mild, 2 = moderate, and 3 = severe). The cumulative scores range from 0 to 12. Two variations of TNSS were considered: r-TNSS, which assesses symptoms throughout the preceding 12 h, and instantaneous TNSS (i-TNSS), which assesses symptoms at the assessment [25]. Participants documented diary entries of r-TNSS and i-TNSS twice a day every day throughout the run-in period, which served as baseline data, as well as the study period, which served as data for confirming changes.

#### 2.6.2. Secondary Outcomes

The Korean Allergic Rhinitis-Specific Quality of Life Questionnaire (KARQLQ) was used to evaluate efficacy on the quality of life [26]. Total IgE and eosinophil counts were examined to assess allergic and inflammatory responses. An overall assessment of AR was performed after treatment [27]. Pattern identification (PI) questionnaire for AR [28] and the Sasang constitution questionnaire [29] were used to provide a comprehensive interpretation based on traditional Korean medicine.

#### 2.6.3. Safety Assessment

Safety was assessed based on vital signs, laboratory examinations, and AEs. Comprehensive laboratory examinations included complete blood cell counts, blood chemistry analyses, urinalysis, serum hepatitis tests, and blood coagulation tests, conducted at screening and during the fourth and fifth visits. The investigator assessed both unexpected and expected AEs and vital signs at every visit and recorded relevant details on a case report form.

### 2.7. Statistical Analysis

Data analysis was conducted by an independent statistician using SAS Analytics Pro. The primary analysis involved a full analysis set (FAS) assessment based on the intention-to-treat approach, whereas the secondary outcome involved per-protocol (PP) analysis. Data are expressed as mean ± standard deviation. Baseline differences between groups were analyzed using analysis of variance for continuous values and Fisher’s exact test or Pearson’s χ^2^ test for categorical values. The outcomes of efficacy (including mean differences in TNSS between the run-in period and the study period) were examined using analysis of covariance, with run-in period data as a covariate in the linear mixed models. That method was also performed for PI, the Sasang constitution, KARQLQ, and total IgE and eosinophil counts. The overall assessment was analyzed using a general linear model.

Patients who received more than one investigational product were subjected to safety evaluations. A comparative analysis of the number of trial-related AEs was conducted using the Kruskal–Wallis test, and between-group comparisons for the proportion of patients who exhibited AEs at least once were conducted using Fisher’s exact test or Pearson’s χ^2^.

## 3. Results

### 3.1. Participants

A total of 201 patients were screened between July 2021 and May 2023. Ultimately, 105 patients were included in this study, with 35 patients each randomly assigned to three groups. Ninety-six patients were excluded owing to screening test results, with the most prevalent reason being failure to meet the inclusion criteria, notably a positive reaction to MAST ≥ class 2. Five patients dropped out following intervention (three and two for consent withdrawal and exclusion criteria, respectively), resulting in the completion of the study by 100 patients. Therefore, 105 and 100 patients were included in the FAS and PP analyses, respectively (Figure 1). No significant differences were observed among groups regarding baseline demographics and clinical characteristics, except for i-TNSS, which showed a difference between the high- and standard-dose BJIGT groups. Therefore, i-TNSS was used with corrected values in statistical analysis (Table 2).

### 3.2. Total Nasal Symptom Score

The baseline r-TNSS values were 6.78 ± 1.49, 6.15 ± 1.36, and 6.51 ± 1.26 in the high-dose BJIGT, standard-dose BJIGT, and placebo groups, respectively. Over the 4-week treatment period (from weeks 1 to 4, the mean r-TNSS values decreased to 4.52 ± 1.99, 3.97 ± 1.84, and 4.47 ± 1.90 in the high-dose BJIGT, standard-dose BJIGT, and placebo groups, respectively **(**Figure 2a)). The mean differences improved in the following order: high-dose BJIGT, standard-dose BJIGT, and placebo groups. However, no statistically significant differences were observed among groups (*p* = 0.4657 [standard-dose BJIGT vs. placebo]; *p* = 0.8502 [high-dose BJIGT vs. placebo]). Exploratory analysis considering the interaction effect in the change from weeks 1–2 (two weeks) to weeks 3–4 (two weeks) revealed a significant decrease in r-TNSS in the high-dose BJIGT group compared with that in the placebo group (*p* = 0.0464) (Figure 3a). An analysis of individual symptoms (rhinorrhea, nasal congestion, nasal itching, and sneezing) indicated that the BJIGT treatment groups exhibited more improvement than the placebo group in most cases. However, no statistically significant differences were observed between groups (Figure 4).

The baseline i-TNSS values were 6.38 ± 1.63, 5.35 ± 1.90, and 5.87 ± 1.59 in the high-dose BJIGT, standard-dose BJIGT, and placebo groups, respectively. Over the 4-week treatment period (from week 1 to 4), the mean i-TNSS values decreased to 4.29 ± 2.03, 3.53 ± 1.75, and 4.06 ± 1.85 in the high-dose BJIGT, standard-dose BJIGT, and placebo groups, respectively (Figure 2b). The mean differences improved in the following order: high-dose BJIGT, standard-dose BJIGT, and placebo groups. However, no statistically significant differences were observed between groups (*p* = 0.5583 [high-dose BJIGT vs. placebo]; *p* = 0.9052 [standard-dose BJIGT vs. placebo]) (Figure 3b).

### 3.3. Korean Allergic Rhinitis-Specific Quality of Life Questionnaire

The KARQLQ was evaluated at the third, fourth, and fifth visits (weeks 0, 2, and 4, respectively). Although it improved in all groups, no statistically significant differences were observed among groups (Figure 5).

### 3.4. Total IgE and Eosinophil Counts

Analysis of the total IgE and eosinophil counts before and after treatment revealed no significant changes within or among groups (Appendix A).

### 3.5. Overall Assessment

The overall assessments post-treatment were 2.09 ± 0.84, 2.21 ± 0.65, and 2.21 ± 0.59 in the high-dose BJIGT, standard-dose BJIGT, and placebo groups, respectively. No statistically significant differences were observed between groups (*p* = 0.7318).

### 3.6. Pattern Identification for AR

Based on PI for AR using traditional-Korean-medicine-perspective classifications, participants were classified into three categories: lung cold type (50 participants, 48%), lung heat type (40 participants, 38%), and spleen deficiency type (15 participants, 14%). Analyzing by intervention group, in the high-dose BJIGT group, 19 (54%), 10 (29%), and 6 (17%) participants exhibited the lung cold type, lung heat type, and spleen deficiency type, respectively. In the standard-dose BJIGT group, 16 (46%), 12 (34%), and 7 (20%) participants exhibited the lung cold type, lung heat type, and spleen deficiency type, respectively. In the placebo group, 18 (51%), 15 (43%), and 2 (6%) participants exhibited the lung heat type, lung cold type, and spleen deficiency type, respectively. The mean difference in the TNSS outcome measurement at the same dose according to PI type was not statistically significant.

### 3.7. Sasang Constitution

All 105 patients were the Tai-eum (greater yin) person type, representing one of the four Sasang constitutional types; therefore, no additional TNSS analysis based on the Sasang constitution was conducted.

### 3.8. Safety Assessment

Regarding AEs, 14 AEs were reported in 10 of the 105 participants in the FAS. The distribution included four, four, and two participants in the high-dose BJIGT, standard-dose BJIGT, and placebo groups, respectively; however, no statistically significant differences were observed among the groups (*p* = 0.7668). Fourteen AEs were considered mild, except for one moderate case of a toe fracture caused by casual activities. All AEs were considered either probably or definitely unrelated to the trial, with confirmation of disappearance post-trial, except for one case of an increase in creatine phosphokinase. Although this instance lacked follow-up, it was attributed to excessive exercise and was deemed clinically insignificant without accompanying symptoms.

Regarding vital signs and laboratory examinations, in most cases, no statistically significant differences were observed between groups. In cases where differences were observed, they fell within the normal range and lacked clinical significance. (Appendix A).

## 4. Discussion

AR is a prevalent condition that significantly affects the daily lives of many individuals, imposing a socioeconomic burden on the public health system [4]. AR often requires long-term treatment, and conventional pharmacological approaches commonly cause AEs [8]. Therefore, there is a growing interest in complementary medicine comprising natural herbs with a low risk of AEs. For example, BJIGT, a widely used herbal medicine with longstanding historical usage, has been used clinically for allergic disease management. Previous investigations have substantiated the anti-inflammatory and immunoregulatory effects of BJIGT in animal models [13,14,30,31], with corroborative evidence of these effects observed in clinical trials involving patients with AR [15,16]. These results support the idea that BJIGT is beneficial for patients with AR, and the above mechanisms are summarized in Figure 6. In the context of traditional medicine, BJIGT is typically used during the chronic stage as preventive therapy targeting recurrent acute attacks. Therefore, we evaluated the efficacy and safety of BJIGT in PAR treatment.

In this study, nasal symptoms improved in all groups, and the mean difference in TNSS from baseline to week 4 in the high-dose BJIGT group was consistent with that of previous studies involving other herbal medicines [11,32]. This consistency suggests clinically significant improvement. However, no statistically significant differences were observed between groups. Although several factors may contribute to these results, a strong placebo effect has been observed in AR when the assessed parameters are subjective [33]. A meta-analysis conducted to evaluate the efficacy of diverse AR treatments in the USA noted high magnitudes of placebo response: 15% improvement in TNSS for seasonal AR and 24.8% improvement for perennial AR [34]. Moreover, the baseline mean r-TNSS was low at approximately 6 points, considering that TNSS ranges up to 12 points, and the inclusion criterion was a minimum of 5 points. This distribution toward the mild grade can hinder the demonstration of differences between groups beyond the placebo effect. Therefore, future studies should contemplate a design targeting moderate and severe AR to obtain definite results that exclude such effects.

In addition, exploratory analysis considering the interaction effect between the groups, severity (mild/moderate/severe), and time (treatment period) revealed a statistically significant decrease in r-TNSS in the high-dose BJIGT group from weeks 1–2 to weeks 3–4 compared with the placebo group, indicative of an interaction of the group*time effect. This suggests that, in the high-dose group rather than in the standard-dose group, an extended treatment period tended to enhance symptom improvement in the BJIGT group compared with the placebo group. Therefore, future studies should incorporate high-dose BJIGT and placebo groups with a larger sample size.

Regarding biomarkers, given the high variability and non-uniform increase in total IgE in all patients with AR and the potential to increase in other disease settings, future studies may benefit from measuring specific IgE levels.

No serious AEs were observed in this trial, and the several mild AEs observed were not clinically significant. Vital signs and laboratory examinations conducted during the trial exhibited no significance among or within groups, indicating the safety of BJIGT during the drug administration period.

This study had several limitations. First, the relatively short treatment period and lack of a follow-up evaluation or additional investigations after the 4-week treatment period constitute a constraint. However, BJIGT is inherently positioned as beneficial for chronic disease management and prevention based on the traditional medicine perspective; therefore, observing the long-lasting effects of treatment over an extended treatment period is imperative. In addition, considering that previous studies investigating BJIGT for PAR showed significant effects over several months of use [15,16], there may be further improvement when BJIGT is taken for more than 4 weeks. This recommendation is underscored by the observed inclination toward improvement as the treatment period lengthened. Second, the absence of adolescent participants is a notable limitation. Only adults were enrolled in this study, overlooking the higher prevalence of AR in school-age children than in adults. Therefore, further research targeting adolescent children should be conducted. Third, objective outcome measures such as nasal mucosal plasma flow and gland exocytosis were absent in this study. Given that the primary outcome measure, TNSS, depends on the subjective recall assessment of patients, it introduces considerable inter-individual variation and is significantly influenced by other variables, such as the placebo effect. To address this limitation, the incorporation of additional objective evaluation tools is deemed necessary.

Despite these limitations, this study holds significance as an exploratory phase II study, offering valuable data for future research endeavors. In future studies, high-dose BJIGT and control groups should incorporate larger sample sizes. Additionally, follow-up evaluation and the targeting of moderate/severe AR are recommended.

## 5. Conclusions

In conclusion, in this clinical trial, we evaluated the efficacy of BJIGT for PAR treatment. Although no statistically significant differences were observed between groups, a notable trend toward improvement was observed, particularly with higher doses and longer treatment periods. Additionally, the safety profile of BJIGT was confirmed. Therefore, this exploratory phase II study offers valuable data for informing future studies in this field.

## Figures and Tables

**Figure 1 healthcare-12-01017-f001:**
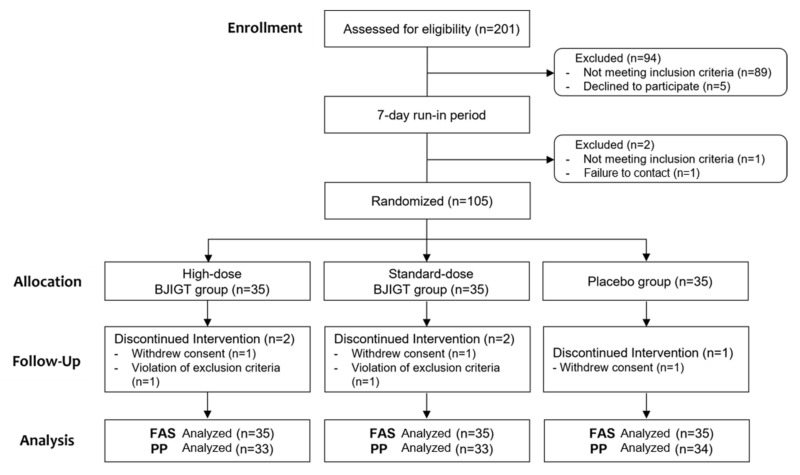
Flow chart of participants.

**Figure 2 healthcare-12-01017-f002:**
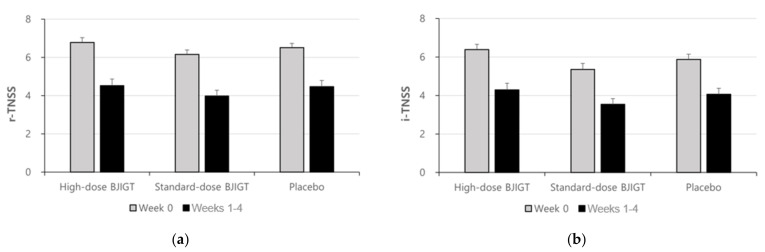
Results for Total Nasal Symptom Score (TNSS): (**a**) reflective-TNSS; (**b**) instantaneous-TNSS. Results are expressed as mean values with standard errors in the full analysis set assessment.

**Figure 3 healthcare-12-01017-f003:**
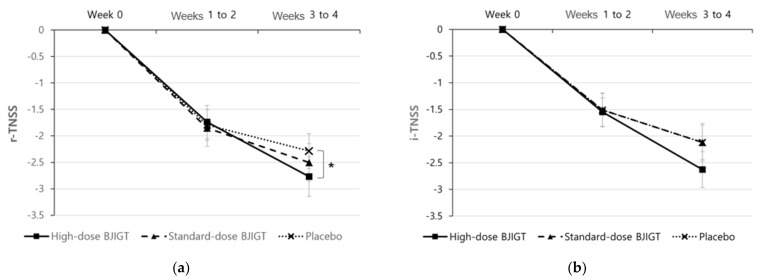
Change from baseline in Total Nasal Symptom Score (TNSS: (**a**) reflective-TNSS (**b**) instantaneous-TNSS. Results are expressed as mean values with standard errors in the full analysis set assessment. * *p* < 0.05: time effect of high-dose Bojungikgi-tang vs. placebo in a linear mixed model, with the interaction of group x time effect.

**Figure 4 healthcare-12-01017-f004:**
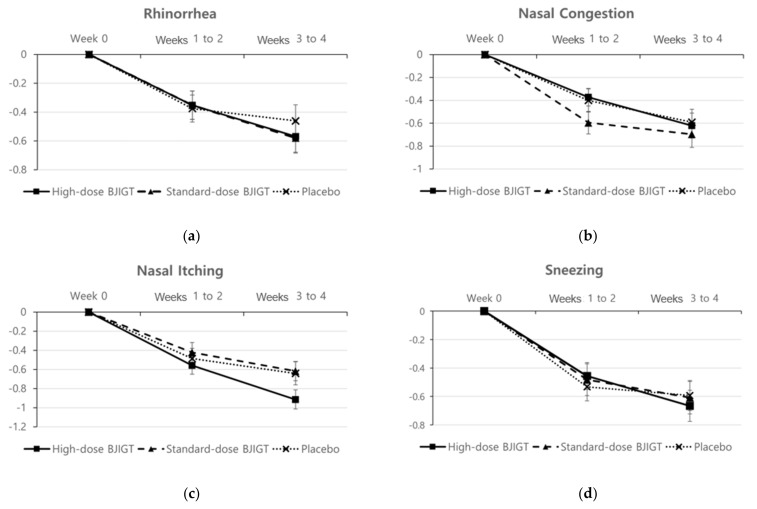
Change from baseline in reflective Total Nasal Symptom Score (r-TNSS) based on symptoms: (**a**) rhinorrhea; (**b**) nasal congestion; (**c**) nasal itching; (**d**) sneezing. Results are expressed as mean values with standard errors in the full analysis set assessment.

**Figure 5 healthcare-12-01017-f005:**
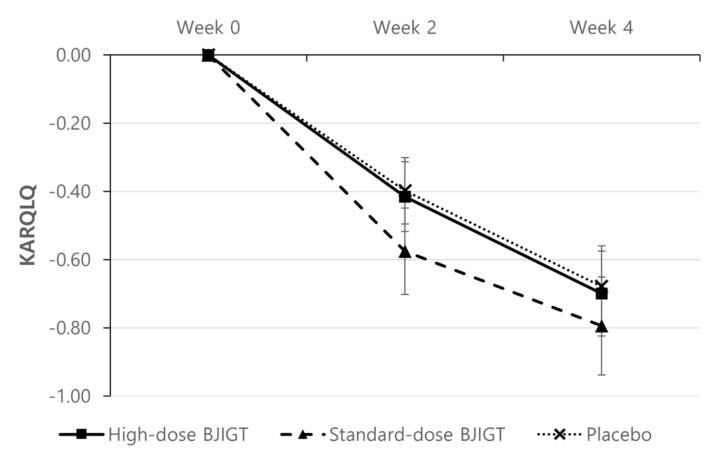
Change from baseline in the Korean Allergic Rhinitis-Specific Quality of Life Questionnaire. Results are expressed as mean values with standard errors in the full assessment set assessment.

**Figure 6 healthcare-12-01017-f006:**
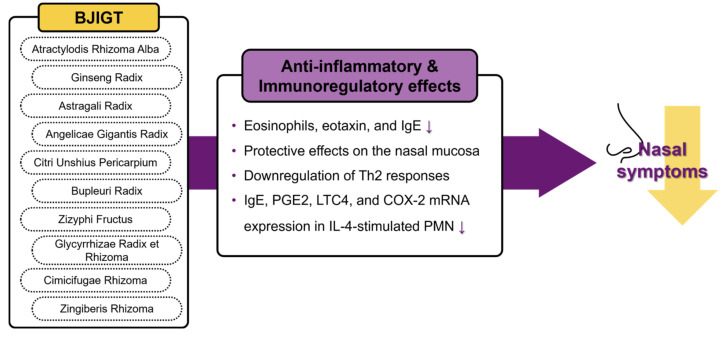
The potential mechanisms of BJIGT for allergic rhinitis. BJIGT, Bojungikgi-tang.

**Table 1 healthcare-12-01017-t001:** Components of Bojungikgi-tang.

Botanical Name	Latin Name	Amount (g)
*Atractylodes macrocephala* Koidzumi	*Atractylodis Rhizoma Alba*	0.665
*Panax ginseng* C.A.Meyer	*Ginseng Radix*	0.665
*Astragalus membranaceus* Bunge	*Astragali Radix*	0.665
*Angelica gigas* Nakai	*Angelicae Gigantis Radix*	0.500
*Citrus reticulata* Blanco	*Citri Unshius Pericarpium*	0.335
*Bupleurum falcatum* Linné	*Bupleuri Radix*	0.335
*Zizyphus jujuba* Miller var. inermis Rehder	*Zizyphi Fructus*	0.335
*Glycyrrhiza uralensis* Fischer	*Glycyrrhizae Radix et Rhizoma*	0.250
*Cimicifuga heracleifolia* Komarov	*Cimicifugae Rhizoma*	0.165
*Zingiber officinale* Roscoe	*Zingiberis Rhizoma*	0.085

**Table 2 healthcare-12-01017-t002:** Baseline characteristics.

	High-Dose BJIGT (*n* = 35)	Standard-Dose BJIGT (*n* = 35)	Placebo (*n* = 35)	*p*-Value
Age (years) ^†^	36.74 ± 12.78	34.74 ± 11.93	33.03 ± 11.47	0.4391
Sex (N, male/female) ^§^	14/21	14/21	14/21	1.000
Weight (kg) ^†^	64.39 ± 11.85	66.58 ± 12.85	67.29 ± 15.29	0.6427
Height (cm) ^†^	164.76 ± 8.71	166.38 ± 7.13	165.67 ± 8.84	0.7133
r-TNSS ^†^	6.78 ± 1.49	6.15 ± 1.36	6.51 ± 1.26	0.1632
i-TNSS ^†^	5.35 ± 1.90	6.38 ± 1.63	5.87 ± 1.59	0.045
KARQLQ ^†^	29.8 ± 10.44	28.86 ± 11.85	28.63 ± 10.78	0.7989
Total IgE ^†^	350.92 ± 621.52	222.75 ± 391.71	256.75 ± 239.64	0.4633
Eosinophil Counts ^†^	205.43 ± 184.44	145.54 ± 99.51	234.86 ± 181.78	0.0639

Values are expressed as mean ± SD. ^†^: one-way ANOVA, ^§^: Fisher’s exact tests KARQLQ; Korean Allergic Rhinitis-Specific Quality of Life Questionnaire; TNSS, Total Nasal Symptom Score; i-TNSS, instantaneous TNSS; r-TNSS, reflective TNSS; IgE, immunoglobulin E; BJIGT, Bojungikgi-tang; SD, standard deviation; ANOVA; analysis of variance.

## Data Availability

The data presented in this study are available upon request from the corresponding authors. The data are not publicly available due to privacy and ethics.

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
