# Peer review of "Efficacy and Safety of Bojungikgi-Tang for Persistent Allergic Rhinitis: A Randomized, Double-Blinded, Placebo-Controlled, Phase II Trial"

_healthcare, 2024, doi:10.3390/healthcare12101017_

Round 1
Reviewer 1 Report
Comments and Suggestions for Authors
There is problem in the evaluation of statistics of this manuscript.
Within text as well as in table 1 the authors claims "A 2.5 g sample of BJIGT contained ten raw medicinal herbs in the following proportions........" if all the components are added, the weight comes out to be 4.765 grams.
In figure 1, the caption should be placed in right position.
In table 2, for Ige the mean values are less than standard deviation. How it can be justified? Usually mean values are high then standard deviation.
In table 2, why Ⴕ, is placed with p-values.
The values regarding figure 2 a within the text are; 4.52±1.99, 3.97±1.84, and 4.47±1.90. These values are not appropriately in the figure (please see the standard deviation).
In figure 4 (d) standard deviation is missing.
The authors claim that the differences in the data in figure 4 are not significant? Where are the p-values? Neither presented in text nor in figure.
Table 3 is confusing. The differences in values should be presented as percent differences in the mean values not as "mean +/- SD" e.g. the eosinophil difference is 0.00±122.45, this is not possible. This further creates doubt on the p-value. In the same table the value 2.22±80.24 is again confusing.
In discussion, "In this study, nasal symptoms improved in all groups (p<0.0001: within-group analysis vs. baseline)," In the results section no such p-value is mentioned. How can the authors claim this value in the discussion.
Comments on the Quality of English Language
Minor changes required
Reviewer 2 Report
Comments and Suggestions for Authors
The manuscript is entitled “Efficacy and Safety of Bojungikgi-tang for Persistent Allergic Rhinitis: A Randomized, Double-blinded, Placebo-controlled, Phase II Trial”. Its purpose was to evaluate its effectiveness against persistent AR (PAR). After a week run-in and 4-week treatment period, patients with PAR were randomly assigned in a 1:1:1 ratio to high-dose BJIGT, standard-dose BJIGT, or placebo groups. The main outcome of the study was the average change in the Total Nasal Symptom Score (TNSS). Secondary outcomes included the Sasang constitution, the Korean Allergic Rhinitis-specific Quality of Life Questionnaire, biomarkers, an overall assessment, and TNSS by AR pattern identification. Despite the fact that the mean TNSS change in the BJIGT group was greater than in the placebo group, the difference was not statistically significant. Further interaction effect analysis demonstrated a statistically significant improvement in the high-dose BJIGT group from weeks 1–2 to weeks 3–4 in comparison to the placebo group. In relation to secondary outcomes, the BJIGT group demonstrated comparable or enhanced results in comparison to the placebo group; these differences lacked statistical significance. There were no notable adverse effects or clinically significant deviations in safety evaluations that were identified. The work is well-written and well-organized. However, there are a few problems that need to be addressed.
In line 140, the detail in reference number 24 is not clear and is hard to understand because it is in Korean. The authors should provide more details.
In discussion, the authors should make some predictions based on the previous studies about what will happen if the patients take the herbal medicine for more than four weeks.
Lines 296–298 in the manuscript or supplementary should summarize biochemical data such as blood chemistry analyses (liver and kidney function tests), urine tests, and serum hepatitis tests.
Please discuss the differences among "Lee, S. W., Choi, J. K., Lyu, Y. R., Yang, W. K., Kim, S. H., Kim, J. H., Kim, S. Y., Kang, W., Jung, I. C., Lee, B. J., Choi, J. Y., Kim, T., & Park, Y. C. (2022). Efficacy and Safety of Bojungikgi-Tang for Persistent Allergic Rhinitis: A Study Protocol for a Randomized, Double-Blind, Placebo-Controlled, Phase II Trial. Evidence-based complementary and alternative medicine : eCAM, 2022, 4414192. https://doi.org/10.1155/2022/4414192” , Lee, S. W., Lyu, Y. R., Kim, S. Y., Yang, W. K., Kim, S. H., Kim, K. M., Chae, S. W., Kang, W., Jung, I. C., & Park, Y. C. (2022). Efficacy and Safety of GHX02 in the Treatment of Acute Bronchitis and Acute Exacerbation of Chronic Bronchitis: A Phase â…¡, Randomized, Double-Blind, Placebo-Controlled, Multicenter Trial. Frontiers in pharmacology, 12, 761575. https://doi.org/10.3389/fphar.2021.761575 and this manuscript.
Reviewer 3 Report
Comments and Suggestions for Authors
I have assessed this manuscript thoroughly and carefully, this manuscript is interesting, but I am not sure whether it is suitable for the journal "Healthcare (ISSN 2227-9032)" I submit this assessment to the editor.
Here are my comments:
1. CONSORT Checklist must be provided as a supplementary file.
2. Biomechanism figures must be provided to make it easier for readers to understand this manuscript.
3. How is the sample size determined? What sample size formula do you use?
Round 2
Reviewer 1 Report
Comments and Suggestions for Authors
The values in table 3, are still confusing, especially the differences. Please move the table to supplementary data. Rest of the questions are answered and justified. The authors have provided the supplementary material as well. The authors are requested to give table number and legends to these tables. Please mention the number of these tables within the manuscript appropriately.
Comments on the Quality of English LanguageMinor changes and careful read is required.
Author Response
The values in table 3, are still confusing, especially the differences. Please move the table to supplementary data. Rest of the questions are answered and justified. The authors have provided the supplementary material as well. The authors are requested to give table number and legends to these tables. Please mention the number of these tables within the manuscript appropriately.
→ We moved table 3 to supplementary materials and organized the supplementary tables.
Reviewer 3 Report
Comments and Suggestions for Authors
The authors revised carefully.
This manuscript could be published in this current form.
Comments on the Quality of English LanguageMost of the typos were revised.
Author Response
Thank you again so much for taking the time to review this manuscript